# Genotype by environment interaction, correlation, AMMI, GGE biplot and cluster analysis for grain yield and other agronomic traits in sorghum (*Sorghum bicolor* L. Moench)

**Muluken Enyew**[1,2]*, **Tileye Feyissa**[1], **Mulatu Geleta**[2], **Kassahun Tesfaye**[1¤], **Cecilia Hammenhag**[2], **Anders S. Carlsson**[2]

**1** Institute of Biotechnology, Addis Ababa University, Addis Ababa, Ethiopia, **2** Department of Plant Breeding, Swedish University of Agricultural Sciences, Alnarp, Sweden

¤ Current address: Ethiopian Biotechnology Institute, Addis Ababa, Ethiopia
* muluken.birara@aau.edu.et, mulukenbi@gmail.com

## Abstract

Genotype by environment (G×E) interaction is a major factor limiting the success of germplasm selection and identification of superior genotypes for use in plant breeding programs. Similar to the case in other crops, G×E complicates the improvement of sorghum, and hence it should be determined and used in decision-making programs. The present study aimed at assessing the G×E interaction, and the correlation between traits for superior sorghum genotypes. Three hundred twenty sorghum landraces and four improved varieties were used in alpha lattice experimental design-based field trial across three environments (Melkassa, Mieso and Mehoni) in Ethiopia. Phenotypic data were collected for days to flowering (DTF), plant height (PH), panicle length (PALH), panicle width (PAWD), panicle weight (PAWT) and grain yield (GY). The results revealed that the variance due to genotype, environment and G×E interaction were highly significant (P < 0.001) for all traits. GY and PAWT were highly affected by environments and G×E whereas DTF, PALH, PAWD and PH were mainly affected by genotypic variation. Therefore, multi-environment testing is needed for taking care of G × E interaction to identify high yielding and stable sorghum landraces. GY and PAWT revealed highly significant positive correlations indicating the possibility of effective selection of the two traits simultaneously. Among the studied populations, South Wello, West Hararghe and Shewa zones had highly diverse genotypes that were distributed across all clusters. Hence, these areas can be considered as hotspots for identifying divergent sorghum landraces that could be used in breeding programs. Melkassa was the most representative environment whereas Mieso was the most discriminating. Five genotypes (G148, G123, G110, G203 and G73) were identified as superior across the test environments for grain yield with farmer-preferred trait, such as plant height. The identified stable and high yielding genotypes are valuable genetic resources that should be used in sorghum breeding programs.

**Data Availability Statement:** All relevant data are within the paper and its Supporting Information files.

**Funding:** This research work was financially supported by the Swedish International Development Cooperation Agency (Sida) Research and Training Grant awarded to the Addis Ababa University and the Swedish University of Agricultural Sciences (AAU-SLU Biotech; https://sida.aau.edu.et/index.php/biotechnology-phd-program/; accessed on September 24, 2021). The funder had no role in the study design; in the data collection, analysis and interpretation; in the writing of the report; and in the decision to submit this manuscript for publication.

**Competing interests:** The authors have declared that no competing interests exist.

# Introduction

Sorghum (*Sorghum bicolor* (L.) Moench) belongs to the grass family *Poaceae* (*Gramineae*). It is a predominantly self-pollinated [1] diploid (2n = 2x = 20) species with a genome size of ca 700 Mbp [2]. Globally, it is the fifth most important cereal crop only surpassed by maize, rice, wheat, and barley, with a global production estimated at 59.3 million metric tonnes (MMT) in 2019/2020 [3,4]. In Africa, sorghum is the second most widely cultivated cereal crop following maize, with a total production of 29.8 MMT on 29.7 million ha of cultivated land [3]. Ethiopia is the world's third largest sorghum producer with a total production of 5.2 MMT following the United States (8.6 MMT) and Nigeria (6.7 MMT) [4]. The national average of its productivity in Ethiopia is 2.69 tha$^{-1}$ [3], which is low when compared to its grain yield potential. However, its grain yield varied from 3.3 to 4.8 tha$^{-1}$ on well-managed fields and experimental plots [5].

Sorghum is a multipurpose crop that is being used for food, feed, and construction as well as in the sugar and molasses industry [6]. It is a major food and nutritional security crop for more than 500 million people in Africa, Asia and Latin America, particularly for those in semi-arid tropical regions, including Ethiopia [7]. Sorghum grows under a wide range of environmental conditions and shows better drought tolerance as compared to other cereal crops. However, the productivity of sorghum is low due to several factors, including limited availability of stable and well-adapted cultivars tolerant to abiotic and biotic stresses.

Adaptability and yield stability are important measures for effective cultivation of a crop species in different agro-climatic regions. The stability and adaptability of genotypes across different environments have been assessed through the application of various statistical tools such as joint regression [8], stability models [9], additive main effects and multiplicative interaction (AMMI) [10], and genotype main effects in addition to genotype by environment interaction (GGE) biplots [11]. AMMI and GGE biplots are the most effective and commonly used multivariate models for the analyses of stability, adaptability and ranking of genotypes and for selecting suitable mega environments [10,12–14]. Both models integrate principal component analysis (PCA) and biplot for the explanation of genotype by environment interaction (G×E). The AMMI model combines analysis of variance (ANOVA) and PCA for the stability analysis of genotypes in a multi-environment trial (MET) dataset [12].

The AMMI stability value (ASV) is derived from the interaction principal component (IPCA1 and IPCA2) scores of the AMMI model [13], which is used to select the most stable genotypes across environments. In AMMI analysis, low ASV indicates high stability of genotypes across environments. However, stable genotypes may not have high mean yield performance. Genotype selection index (GSI) was developed for selection of the best genotype, which has both high mean performance and stability. Low GSI values indicate high mean performance and stability of genotypes [15,16]. The GGE biplot combines two important sources of variation in MET (Genotype and G×E). It is used for mega environment analysis ("Which-Won-Where" pattern), evaluation of genotype (ranking biplot) and environment (comparison biplot), which provides discriminating power and representation of the environments [14,16].

The majority (85%) of the improved sorghum varieties released for use in the lowland and mid-altitude environments in Ethiopia were developed based on exotic germplasm [17,18]. However, the released varieties had very low adaptation rates, as they lack farmers preferred traits such as grain quality, grain size, and biomass. The biomass, which is used for animal feed, fuel, and construction of fences is often valued as high as grain yield [18,19] and taller varieties are highly favored by farmers. Selecting sorghum landraces possessing the preferred traits of plant height and grain yield is crucial for farmers' direct use as well as for future sorghum breeding programs. Most studies on sorghum landraces have focused on a relatively

simple evaluation of grain yield, and less effort has been given to advanced and more informative analyses of traits using MET data. Although G×E interaction has been performed to assess the stability of improved varieties of sorghum using MET data [5,20–23], information is not available on the stability of sorghum landraces through the application of AMMI and GGE biplot models. Therefore, the objectives of this study were to evaluate G×E interaction, the performance and stability of sorghum landraces, correlation of grain yield and agronomic traits and to determine representativeness and discriminating ability of different environments where sorghum is cultivated.

## Materials and methods

### Plant materials

In this study, 324 sorghum landrace accessions (320) and improved varieties (4) grown in Ethiopia were used (S1 Table). Among the 320 landrace accessions, 261 were obtained from Melkassa Agricultural Research Center (MARC), but originally collected by Ethiopian Biodiversity Institute (EBI), whereas 59 accessions were newly collected from farmers' fields in drought prone areas. The four improved varieties (Melkam, Argiti, ESH4 and B35) were obtained from MARC. Hereafter, both landrace accessions and varieties are referred to as "genotypes" for the sake of simplicity.

### Study locations

This research was carried out in three locations in Ethiopia, namely Melkassa (MK), Mieso (MS) and Mehoni (MH), during the main crop growing season in 2019 (Table 1, Fig 1). These sites represent moisture stress areas in the country where sorghum is predominantly grown by smallholders.

### Experimental design and field managements

The experiment was laid out as a 27 × 12 alpha lattice design with two replications across three environments. Each plot had an area of 2.25 m$^2$ (3 m × 0.75 m) and seeds were sown in a single 3 m long row on each plot. Planting was done manually followed by thinning to 0.20 m space between plants. The recommended amount of DAP fertilizer (100 kgha$^{-1}$) was applied during planting and urea (50 kgha$^{-1}$) was side dressed 40 days after planting. All necessary agronomic practices were applied following standard procedures for sorghum (maybe a reference here also).

### Collecting phenotypic data

All phenotypic data were collected from five randomly selected and tagged plants in each plot. Days to flowering (DTF) were recorded as the number of days from planting to flowering of 50% of the plants on a plot. Panicle length (PALH) and panicle width (PAWD) were measured as the length of the panicle from the base to the tip of the panicle and as the width of the panicle at its widest section, respectively. Plant height (PH) was measured as the height of the plant from the base to the tip of a panicle at maturity. Grain yield (GY) was recorded as the weight

**Table 1. Description of the testing environments.**

| ENV | Distance from AA (km) | Region | District | Annual rainfall (mm) | Soil type | Min–Max T$^O$ | Longitude | Latitude | Altitude m.a.s.l |
|---|---|---|---|---|---|---|---|---|---|
| **Melkassa** | 117 | Oromia | Adama | 763 | Andosol | 14–28.4˚C | 39˚21'E | 8˚24'N | 1550 |
| **Mieso** | 297 | Oromia | Mieso | 570 | Vertisol | 14–34˚C | 7˚31'E | 12˚9'N | 1470 |
| **Mehoni** | 807 | Tigray | Raya Azebo | 750 | Aluvisols | 18–25˚C | 39˚37'E | 8˚41'N | 1574 |

ENV = environment; AA = Addis Ababa; m.a.s.l = meter above sea level; T$^O$ = Temperature.

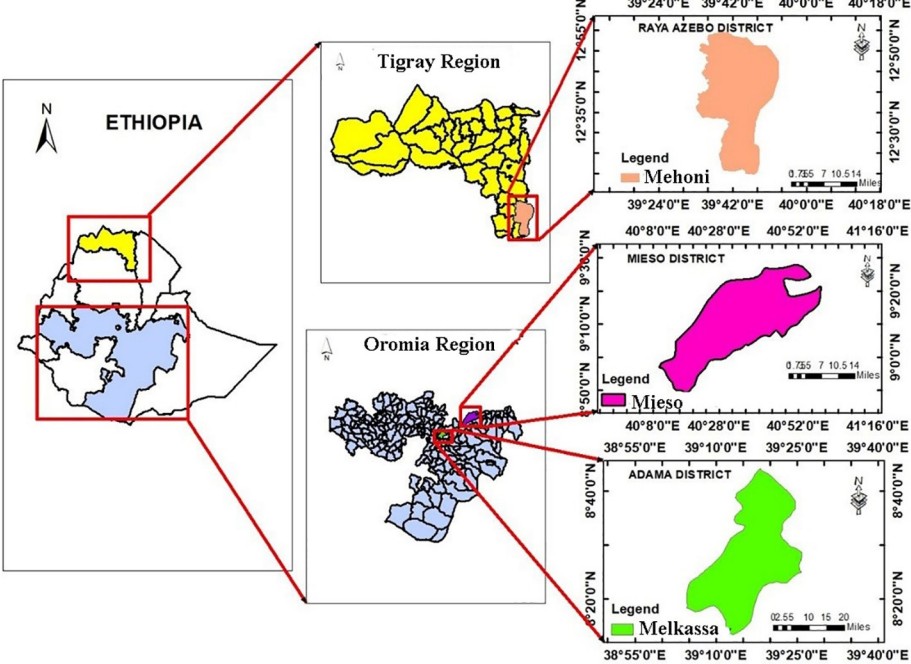

**Fig 1. Geographical map, constructed using geographic information system (ArcGIS), [24] showing the three testing environments: Mehoni, Mieso and Melkassa in Ethiopia.** Mehoni is located in Raya Azebo district of Tigray Regional State whereas Mieso and Melkassa are located in Mieso and Adama districts of Oromia regional State, respectively.

of seeds from an individual plant's panicle whereas panicle weight (PAWT) was measured as the weight of the un-threshed panicle.

## Data analysis

The phenotypic data collected from the three environments were subjected to a combined ANOVA using mixed linear model in R software [25]. The significance level of genotype, environment and G×E interaction effects were then determined. AMMI model was used to determine the G×E interaction effect, assess adaptability and stability of the sorghum landrace across the three environments. The ASV was calculated as described in Purchase et al. [26] to measure and rank the sorghum genotypes based on their stability. GSI was also calculated as described in Purchase et al. [26] using R software. Singular value decomposition (SVD) of the first two principal components was used to fit the GGE biplot model [27]. The AMMI and GGE biplot analyses were done using GENSTAT software [28]. For correlation analysis, BLUPs (Best Linear Unbiased Predictors) were calculated for all traits across the three environments using META-R software [29] and the Pearson correlation coefficient and graphs were generated in R software. Cluster analysis was performed using DendroUPGMA [30], and the tree generated was visualized using MEGA X [31].

## Results

### Combined analysis of variance

The combined ANOVA showed a significant variation for genotype, environment and G×E interaction for all traits studied (P < 0.001) (Table 2). High variability exists in sorghum

**Table 2. Combined analysis of variance for grain yield and related traits of 324 sorghum genotypes across three environments.**

| SOURCE | DF | DTF | | PH | | PAWD | |
|---|---|---|---|---|---|---|---|
| | | SS | MS | SS | MS | SS | MS |
| GEN | 323 | 331,967 | 1028.0*** | 4,177,637 | 1,2933.9*** | 19,502.2 | 60.38*** |
| ENV | 2 | 165,081 | 82541.0*** | 45,146 | 22,573.1*** | 696.4 | 348.18*** |
| REP:ENV | 3 | 765 | 255.0*** | 45,146 | 22,573.1* | 331.7 | 110.57*** |
| GEN:ENV | 646 | 78,630 | 122.0*** | 12,540 | 4,180.1*** | 7,567.9 | 11.71*** |
| BLK:ENV:REP | 66 | 2,218 | 34 | 1,334,826 | 2,066.3*** | 666.2 | 10.09 |
| Residuals | 903 | 27,329 | 30 | 151,408 | 2,294.1 | 7,723.4 | 8.55 |
| Mean | 108 | | | 272.1 | | 9.63 | |
| | DF | PALH | | PAWT | | GY | |
| | | SS | MS | SS | MS | SS | MS |
| GEN | 323 | 95,283 | 295.0*** | 1,416,894 | 4,387.0*** | 890,878 | 2,758.0*** |
| ENV | 2 | 12,224 | 6112.2*** | 2,042,434 | 1,021,217.0*** | 1,000,399 | 500,200.0*** |
| REP:ENV | 3 | 457 | 152.3*** | 5,292 | 1764 | 16,682 | 5,561.0*** |
| GEN:ENV | 646 | 19,208 | 29.7*** | 1,708,715 | 2,645.0*** | 1,119,807 | 1,733.0*** |
| BLK:ENV:REP | 66 | 1,068 | 16.2 | 72,830 | 1,103 | 44,639 | 676 |
| Residuals | 903 | 15,988 | 17.7 | 897,730 | 994 | 579,407 | 642 |
| Mean | 21.3 | | | 104.5 | | 78.14 | |

*** = Significant at 0.001 significance level

** = Significant at 0.01 significance level

* = Significant at 0.05 significance level

DF = Degrees of freedom; GEN = Genotype; REP = Replication; ENV = Environment; BLK = Block; SS = Sum of squares; MS = Mean square, DTF = Days to flowering; PH = Plant height; PALH = Panicle length; PAWD = Panicle width; PAWT = Panicle weight; GY = Grain yield.

genotypes for yield and agronomic traits (Table 2 and Fig 2). The grand mean values were 108 days for DTF, 272.1 cm for PH, 21.3 cm for PALH, 9.6 cm for PAWD, 104.5 g for PAWT and 78.1 g for GY across the three environments (Table 2).

## AMMI analysis of variance

The result of AMMI ANOVA (Table 3) showed that the genotype, environment and G×E interaction effects were highly significant (P < 0.001) for DTF, GY, PH, PALH, PAWD and PAWT. The genotype explained over 50% of the total variation in DTF, PH, PALH and PAWD. In the case of DTF, genotypic variance accounted for 54.9% of the total variance whilst environment and G×E interaction contributed 27.3% and 13.0% to the total variance, respectively. The proportion of the total variance explained by genotype, environment and G×E interaction for PALH were 66.3%, 8.5% and 13.4%, respectively. For PAWD, genotype, G×E interaction and environment effects explained 53.9%, 20.9% and 1.9% of the total phenotypic variance, respectively. For PH the genotype, environment and G×E interaction effects accounted for 61.4%, 0.7% and 19.6% of the variation, respectively. In the case of PAWT, genotype, environment and G×E interaction explained 22.3%, 33.7% and 28.1% of the total variance, in that order. For GY, G×E interaction explained 31.3% of the total variance whereas genotype and environment effects contributed 23.1% and 28.0% to the total variance, respectively.

## Genotype by environment interaction

Based on the AMMI analysis, the mean values of each trait in each environment—the IPCA1 and IPCA2 scores—and the four top ranking genotypes for each trait at each environment are

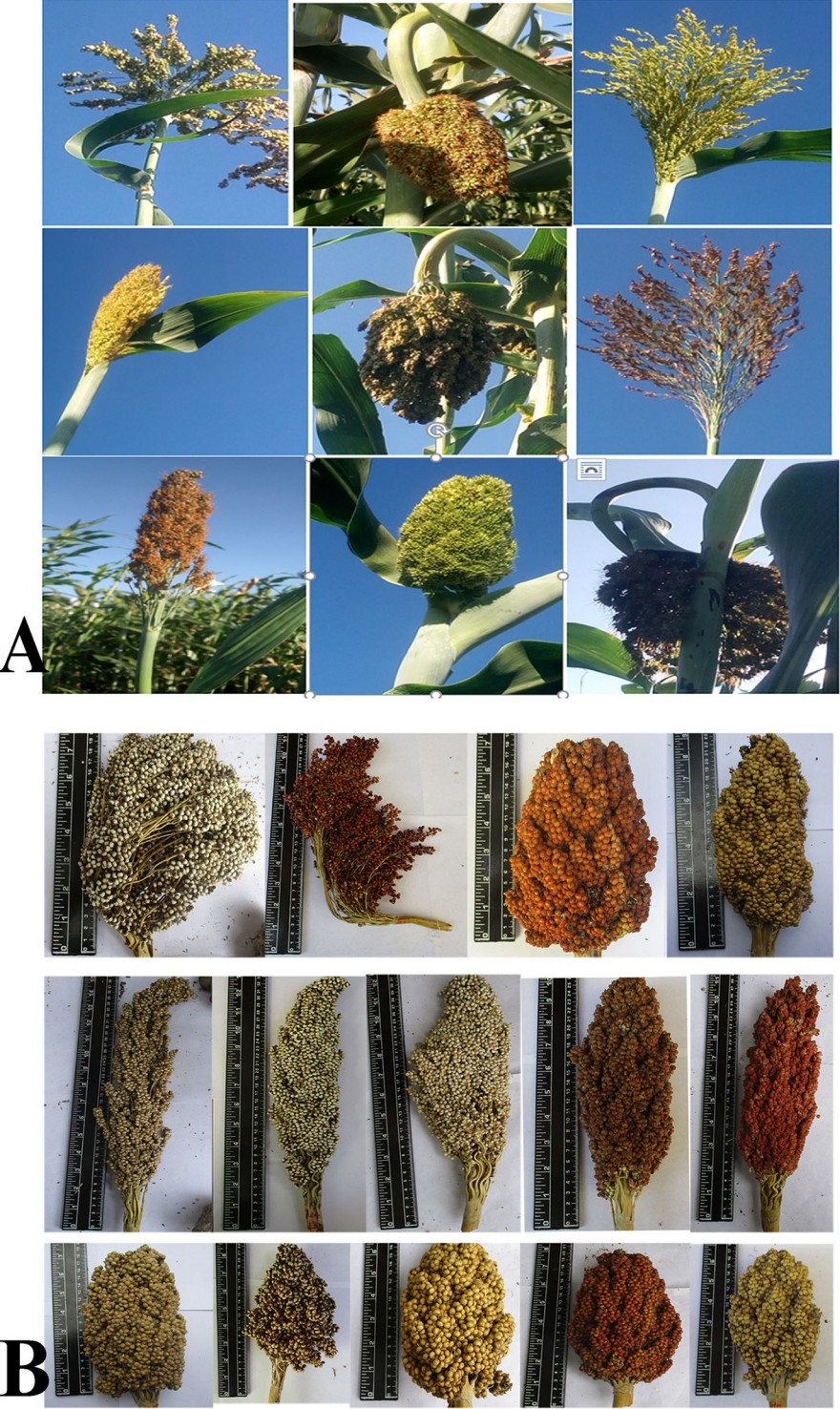

**Fig 2. Diverse sorghum panicles A) at early grain filling and B) at maturity.**

presented in Table 4. Low IPCA1 scores shows low contribution to the G×E interaction and high contribution to genotype stability [32]. In this study, environments contributed differently to the genotype stability for different traits. The IPCA1 scores indicated that Melkassa

**Table 3. AMMI ANOVA for grain yield and related traits of 324 sorghum genotypes across three environments.**

| Source | DF | DTF | | | GY | | | PH | | |
|---|---|---|---|---|---|---|---|---|---|---|
| | | SS | MS | %TV | SS | MS | % TV | SS | MS | %TV |
| **Total** | 1943 | 605,990 | 312 | | 3,550,046 | 1,827 | | 6,815,820 | 3,508 | |
| **Trt** | 971 | 575,678 | 593*** | | 2,909,507 | 2,996*** | | 5,557,609 | 5,724*** | |
| **BLK** | 3 | 765 | 255*** | | 16,536 | 5,512*** | | 12,540 | 4,180** | |
| **GEN** | 323 | 331,967 | 1,028*** | 54.9 | 815,281 | 2,524*** | 23.1 | 4,177,637 | 12,934*** | 61.4 |
| **ENV** | 2 | 165,081 | 82,541*** | 27.3 | 989,114 | 494,557*** | 28 | 45,146 | 22,573*** | 0.7 |
| **G×E** | 646 | 78,630 | 122*** | 13.0 | 1,105,113 | 1,711*** | 31.3 | 1,334,826 | 2,066*** | 19.6 |
| **IPCA1** | 324 | 77,345 | 239*** | 98.4 | 612,196 | 1,889*** | 55.4 | 769,574 | 2,375*** | 57.7 |
| **IPCA2** | 322 | 1,285 | 4 | 1.6 | 492,917 | 1,531*** | 44.6 | 565,252 | 1,755*** | 42.3 |
| **Error** | 969 | 29,547 | 30 | 4.9 | 624,002 | 644*** | 17.7 | 1,245,671 | 1,286 | 18.3 |
| **Source** | DF | PALH | | | PAWD | | | PAWT | | |
| | | SS | MS | %TV | SS | MS | %TV | SS | MS | %TV |
| **Total** | 1,943 | 144,228 | 74.2 | | 36,488 | 18.8 | | 6,050,804 | 3,114 | |
| **Trt** | 971 | 126,716 | 130.5*** | | 27,766 | 28.6*** | | 5,075,294 | 5,227*** | |
| **BLK** | 3 | 457 | 152.3*** | | 332 | 110.6*** | | 1,758 | 1.76 | |
| **GEN** | 323 | 95,283 | 295*** | 66.3 | 19,502 | 60.4*** | 53.9 | 1,347,522 | 4,172*** | 22.3 |
| **ENV** | 2 | 12,224 | 6112.2*** | 8.5 | 696 | 348.2** | 1.9 | 2,028,639 | 1,014,319*** | 33.7 |
| **G×E** | 646 | 19,208 | 29.7*** | 13.4 | 7,568 | 11.7*** | 20.9 | 1,699,133 | 2,630*** | 28.1 |
| **IPCA1** | 324 | 11,786 | 36.4*** | 61.4 | 4,978 | 15.4*** | 65.8 | 994,512 | 3,069*** | 58.5 |
| **IPCA2** | 322 | 7,423 | 23.1** | 38.6 | 2,590 | 8.04 | 34.2 | 704,621 | 2,188*** | 41.5 |
| **Error** | 969 | 17,055 | 17.6 | 11.9 | 8,390 | 8.66 | 23.2 | 970,237 | 1,001*** | 16.1 |

\*\*\* = Significant at 0.001 significance level

\*\* = Significant at 0.01 significance level

\* = Significant at 0.05 significance level

DF = Degree of Freedom; Trt = Treatment; GEN = Genotype; REP = Replication; ENV = Environment; BLK = Block; SS = Sum of square; MS = Mean square; % TV = Percentage of total variance explained; G×E = Genotype by environment interaction; IPCA = Interaction principal component axis; DTF = Days to flowering; PH = Plant height; PALH = Panicle length; PAWD = Panicle width; PAWT = Panicle weight; GY = Grain yield; Source = Source of variation.

(MK) was a main contributor to the stability of genotypes in terms of panicle length (PALH) and width (PAWD). On the other hand, Mehoni (MH) contributed the most to genotype stability in grain yield (GY) and panicle weight (PAWT).

The AMMI2 biplot revealed environment scores with IPCA1 and IPCA2 for grain yield, panicle weight, plant height, panicle length and width (Figs 3 and S1). In the AMMI2 biplot, environments with low IPCA1 and IPCA2 scores that are placed close to the origin have high contribution to the stability of genotypes and low contribution to GE interaction. In this study, AMMI2 biplots indicated that all environments were positioned far from the biplot origin for grain yield, panicle weight, plant height, panicle length and width.

## Genotype performance and AMMI stability analysis

Genotype performance and AMMI stability analysis were conducted and the top and bottom ranking genotypes based on their mean values (Table 5) and genotype selection index (Table 6) are presented. Analysis of the AMMI indicated that genotypes G306, G239, G313, G201, and G213 had high mean grain yield of 150.2, 1363, 133.8, 133.5 and 131.2 g, respectively, while G142, G168 and G321 were the least in grain yield as well as in panicle weight. The high yielding genotypes, G239 and G306 had higher panicle weight, 176.7 and 174.5 g respectively. With regard to panicle length, G244 (39.6 cm) and G118 (38.4 cm) had longest

**Table 4. AMMI analysis based mean phenotypic and IPCAe values, and four top ranking genotypes for each trait in each environment.**

| Traits | ENV | Mean | IPCAe1 | IPCAe2 | 1 | 2 | 3 | 4 |
|--------|-----|------|--------|--------|---|---|---|---|
| **PH** | MH | 273.6 | -20.3 | 0.8 | G255 | G263 | G153 | G210 |
| | MS | 277.2 | 9.4 | -16.7 | G23 | G228 | G145 | G244 |
| | MK | 265.6 | 10.9 | 15.9 | G255 | G307 | G261 | G248 |
| **PALH** | MH | 20.0 | 6.1 | 3.4 | G244 | G143 | G65 | G26 |
| | MS | 19.0 | -6.3 | 3.0 | G244 | G139 | G118 | G271 |
| | MK | 24.8 | 0.3 | -6.4 | G41 | G118 | G149 | G132 |
| **PAWD** | MH | 10.0 | -5.4 | -1.7 | G255 | G12 | G313 | G210 |
| | MS | 8.8 | 4.4 | -3.1 | G5 | G7 | G224 | G261 |
| | MK | 10.1 | 1.0 | 4.8 | G12 | G79 | G307 | G318 |
| **PAWT** | MH | 109.0 | 3.4 | -19.6 | G207 | G305 | G191 | G183 |
| | MS | 62.8 | 16.9 | 12.5 | G163 | G30 | G20 | G273 |
| | MK | 141.5 | -20.2 | 7.1 | G213 | G186 | G97 | G193 |
| **GY** | MH | 82.3 | -1.8 | 18.1 | G207 | G306 | G183 | G313 |
| | MS | 48.2 | 17.5 | -7.5 | G20 | G163 | G226 | G30 |
| | MK | 102.9 | -15.6 | -10.5 | G213 | G306 | G201 | G313 |

ENV = Environment; IPCAe1 and IPCAe2 = The first and the second interaction principal component axis score of environments; PH = Plant height, PALH = Panicle length, PAWD = Panicle width, PAWT: Panicle weight, GY: Grain yield; MH = Mehoni; MS = Mieso; MK = Melkassa.

panicle whereas G93 had shortest panicle, 9.9 cm. G12 (21.8 cm) followed by G255 (20.3 cm) had widest panicle while G157 (5.6 cm) were the least. Similarly, G255 (382.8 cm) followed by G244 (372.1 cm) were the tallest whereas G321 and G322 were the shortest genotypes, 90 cm and 133 cm respectively (Table 5).

Low AMMI stability value (ASV) indicates high stability of genotypes and low G×E interaction [26]. Genotypes G70, G162 and G254 with mean grain yield of 64.8, 68.8, and 66.6 g, respectively, showed high stability having low ASV (S2 Table), but not high yield, and therefore should not be selected. On the other hand, the following genotypes were identified as having high stability and grain yield based on their genotype selection index (GSI): G148, G123, G110, G203 and G73 (Table 6). Among these genotypes, G148 and G73 had high stability and panicle weight. Genotypes G213, G306 and G201 had high mean grain yield but were placed far from the biplot origin suggesting that they were not stable (Table 5). These genotypes appeared to be specifically adapted to environment MK. The positive interaction of G207, G306, G183 and G313 with environment MH, and G20, G163, G226 and G30 with environment MS indicated the specific adaptation of the genotypes for grain yield to the respective environment (Table 4, Fig 3B).

## GGE biplot analysis

**Which-won-where polygon view of GGE biplot.** The polygon view of GGE biplot showed the interaction patterns between genotypes and environments and visualized the best performing genotypes (Figs 4 and S2). In this GGE biplot, a polygon was drawn by joining the vertex genotypes, which were placed far from the origin, with red straight lines and hence, all the other genotypes were enclosed within the polygon. The vertex genotypes for grain yield were G163, G306, G313, G194 G168, G142, G209 and G20. Whereas, genotypes G163, G306, G313, G194, G168, G142, G262 and G20 took the vertices for panicle weight (Fig 4). Hence, these two sets of genotypes were the most responsive to environmental interactions for grain yield and panicle weight in that order. The most responsive genotypes forming the vertices of

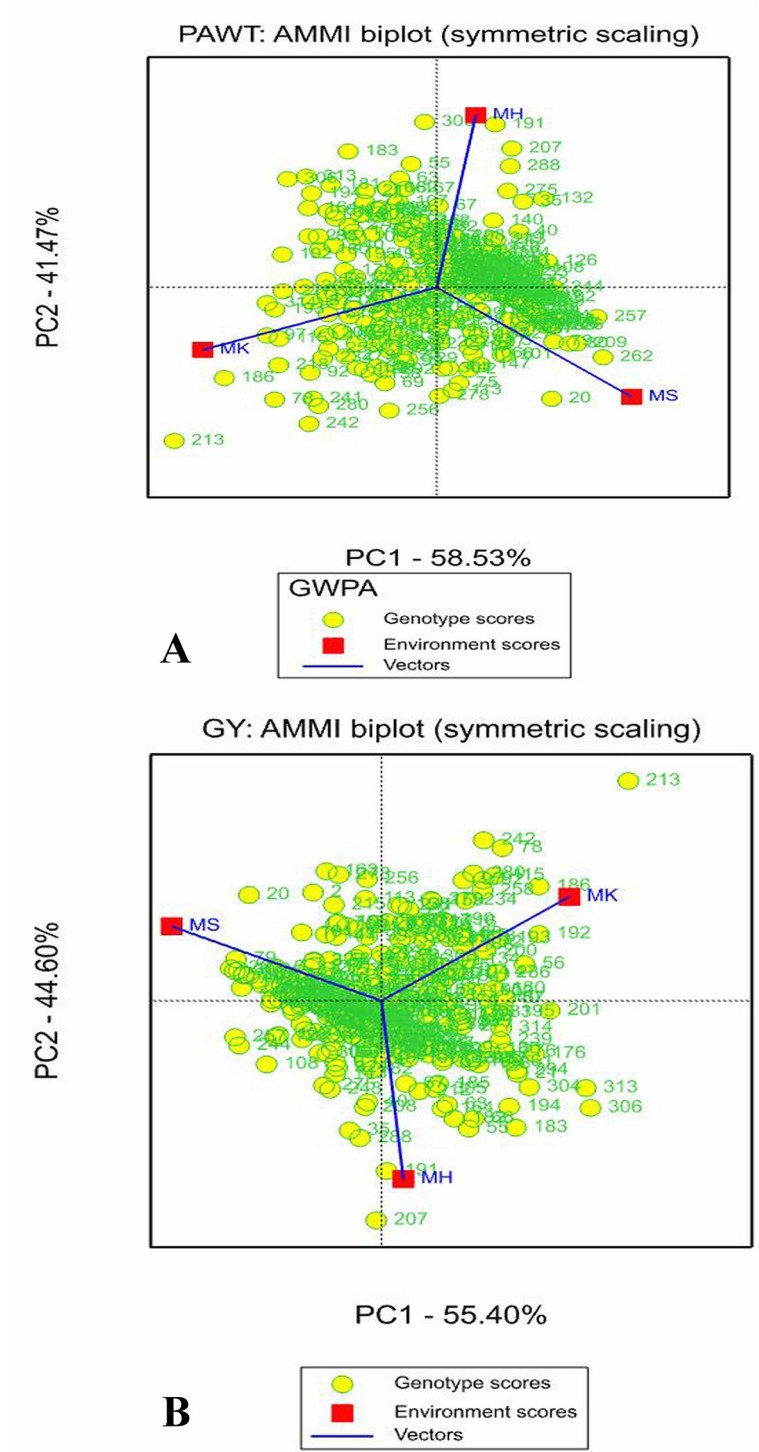

**Fig 3. AMMI2 biplot of the 324 sorghum genotypes and three environments for (A) panicle weight (PAWT) and (B) grain yield (GY).** Genotypes placed close to a given environment, had top performance in that environment. Each vector shows the discrimination power of the environment (the longer the vector the more discrimination power that environment has).

**Table 5. Mean performance of the top ten and bottom five sorghum genotypes across the three environments for grain yield and other agronomic traits.**

| PH (cm) | | PALH (cm) | | PAWD (cm) | | PAWT (g) | | GY (g) | |
|---|---|---|---|---|---|---|---|---|---|
| Gen | Means | Gen | Means | Gen | Means | Gen | Means | Gen | Means |
| G255 | 382.8 | G244 | 39.6 | G12 | 21.8 | G163 | 191.9 | G306 | 150.2 |
| G244 | 372.1 | G118 | 38.4 | G255 | 20.3 | G239 | 176.7 | G239 | 136.3 |
| G261 | 370.8 | G149 | 37.8 | G5 | 19.2 | G164 | 175.4 | G313 | 133.8 |
| G307 | 370.1 | G79 | 36.3 | G210 | 19.1 | G306 | 174.5 | G201 | 133.5 |
| G5 | 367.6 | G248 | 34.8 | G261 | 19.0 | G213 | 168.1 | G213 | 131.2 |
| G319 | 365.1 | G132 | 34.7 | G79 | 19.0 | G105 | 165.1 | G207 | 126.3 |
| G228 | 363.0 | G268 | 34.7 | G248 | 18.8 | G193 | 163.8 | G105 | 125.5 |
| G210 | 360.6 | G41 | 34.5 | G267 | 18.6 | G97 | 160.8 | G119 | 122.7 |
| G263 | 360.6 | G139 | 33.9 | G251 | 18.4 | G201 | 159.0 | G183 | 122.3 |
| G267 | 360.5 | G26 | 33.8 | G252 | 18.1 | G119 | 158.4 | G55 | 118.5 |
| G227 | 154.2 | G304 | 10.4 | G46 | 5.7 | G255 | 56.5 | G8 | 40.5 |
| G46 | 151.8 | G280 | 10.1 | G129 | 5.7 | G94 | 53.0 | G265 | 38.8 |
| G122 | 151.7 | G279 | 10.0 | G199 | 5.7 | G321 | 52.6 | G321 | 38.6 |
| G322 | 133.3 | G93 | 9.9 | G157 | 5.6 | G168 | 47.3 | G168 | 38.4 |
| G321 | 90.8 | G178 | 9.9 | G28 | 5.5 | G142 | 43.9 | G142 | 29.2 |

Gen = Genotype, PH = Plant height, PALH = Panicle length, PAWD = Panicle width, PAWT: Panicle weight, GY: Grain yield. **Note**: The first ten accessions in each "Gen" columns are the top ten whereas the last five accessions are the bottom five for the corresponding traits.

the polygons were G169, G314 and G313 for panicle length, G307, G12, G313 and G5 for panicle width, and G249, G255, G153, G321 for plant height (S2 Fig).

In "which-won-where" GGE biplot, lines from the origin divide the biplot into different sectors and create different mega environments (MGEs) [14,33]. In this study, two MGEs were

**Table 6. The ASV, GSI and combined mean performance of the top ten and bottom five genotypes for grain yield and panicle weight.**

| PAWT | | | | | GY | | | | |
|---|---|---|---|---|---|---|---|---|---|
| Gen | Means | IPCAg1 | ASV | GSI | Gen | Means | IPCAg1 | ASV | GSI |
| G119 | 158.4 | -0.6 | 1.0 | 53.0 | G148 | 96.9 | -0.3 | 0.4 | 76.0 |
| G73 | 136.0 | 0.4 | 0.6 | 55.0 | G123 | 97.6 | -0.4 | 0.5 | 79.5 |
| G219 | 135.5 | 0.2 | 0.6 | 59.0 | G110 | 106.4 | -0.1 | 0.8 | 81.0 |
| G86 | 149.8 | 0.6 | 1.0 | 65.0 | G203 | 99.0 | -0.5 | 0.7 | 88.0 |
| G95 | 133.6 | 0.3 | 0.9 | 80.0 | G73 | 105.5 | -0.1 | 0.9 | 95.0 |
| G148 | 125.1 | -0.2 | 0.7 | 98.0 | G151 | 94.7 | 0.3 | 0.6 | 98.0 |
| G253 | 121.6 | 0.5 | 0.7 | 109.0 | G86 | 103.0 | 0.7 | 0.9 | 108.0 |
| G312 | 132.3 | -0.1 | 1.1 | 110.0 | G269 | 105.6 | -0.4 | 1.0 | 110.0 |
| G163 | 191.9 | 1.1 | 1.5 | 112.0 | G189 | 85.4 | -0.3 | 0.5 | 124.0 |
| G296 | 131.5 | -0.1 | 1.1 | 130.0 | G211 | 86.0 | -0.4 | 0.5 | 124.0 |
| G12 | 80.1 | 2.3 | 3.4 | 542.0 | G229 | 42.0 | 1.6 | 2.1 | 539.0 |
| G265 | 57.4 | 1.9 | 2.7 | 543.0 | G318 | 59.9 | 2.4 | 3.0 | 549.0 |
| G209 | 83.5 | 2.8 | 4.2 | 547.0 | G249 | 48.4 | 2.2 | 2.7 | 576.0 |
| G7 | 78.2 | 2.3 | 3.4 | 551.0 | G8 | 40.5 | 2.0 | 2.5 | 579.0 |
| G262 | 80.4 | 3.3 | 5.0 | 578.0 | G209 | 49.9 | 2.6 | 3.3 | 597.0 |

Gen = Genotype, ASV: AMMI stability value, GSI: Genotype selection index, IPCAg1: First interaction principal component axis scores for genotype, PAWT: Panicle weight, GY: Grain yield. **Note**: The first ten accessions in each "Gen" columns are the top ten whereas the last five accessions are the bottom five for the corresponding traits.

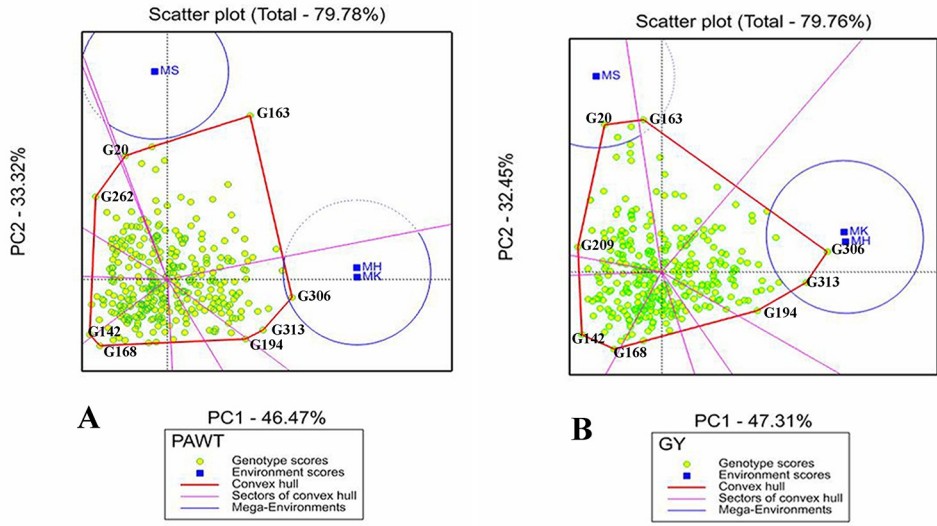

**Fig 4. Which-Won-Where polygon view of GGE scatter biplot of the 324 sorghum genotypes for (A) Panicle weight (PAWT) and (B) Grain yield (GY), showing genotypes with best performance in each environment and the mega environments (MGEs).** The vertex genotypes on convex hull (Polygon) are the best in each mega environment for the corresponding trait.

formed for all traits except for panicle length that had three MGEs. For grain yield, panicle weight and panicle width, environments MK and MH jointly formed a MGE whereas environment MS was a separate MGE for these traits. On the other hand, environments MS and MK jointly formed an MGE for plant height with MH forming a separate MGE (Figs 4 and S2). Inside the sector containing the first mega environment for grain yield and panicle weight, genotypes at the vertices of the polygons were G306, G313 and G194 indicating that they are top performers in the environment MK and MH. In the GGE biplot analysis, the partitioning of GE interaction revealed that the first two PCs contributed 79.78%, 79.76%, 94.17%, 89.19% and 89.98% of the total variation in panicle weight, grain yield, panicle length and width, and plant height, respectively (Figs 4 and S2).

**Genotype ranking based on their mean performance and stability.** Ranking biplots were used to rank the genotypes according to their performance and stability using the average environment coordinate (AEC) [13]. An average environment axis (AEA) in the ranking biplot represented by a single arrowhead line that passes through the origin shows higher mean performance of a genotype. In this study, the ranking biplot AEC showed that genotypes G306, G239, G201, G213, G207 and G105 had high mean GY and genotypes G163, G239, G164, G105 and G213 had high mean PAWT. On the other hand, genotypes G321, G168 and G142 had the lowest grain and panicle weight in that order (Fig 5). In PALH, genotypes G244, G118 and G149 came out on top whereas G12, G255 and G5 were the top ranking in PAWD. Genotypes with the shortest panicle were G178, G93 and G279 whereas G28, G157 and G199 were the bottom ranking in PAWD. In plant height, G255, G244 and G261 were the top ranking whereas G121, G322 and G32 where the shortest genotypes (S3 Fig).

The stability of genotypes were evaluated based on the length of the vector (dotted line in the graph) between the genotype positions and the AEA in ranking biplot (Figs 5 and S3). The best performing and stable genotypes are those that are far from the origin but on the AEA or close to it. Hence, G119, G105, G213, G239 and G207 were the most stable genotypes with high mean GY that had shorter vector from AEA whereas G20 and G163 were the least stable genotypes having longest vector from AEA (Fig 5B). For PAWT, G207 and G213 were the

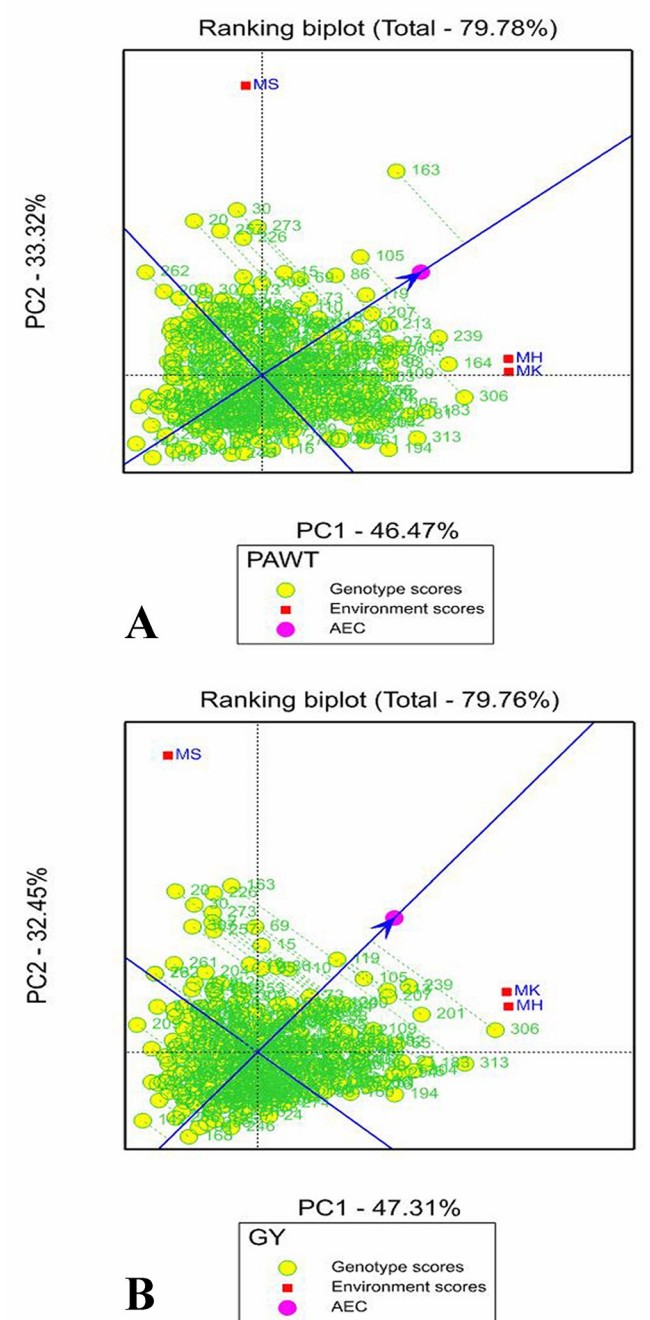

**Fig 5. Genotype focus scaling of GGE biplot showing stability and mean ranking of the 324 sorghum genotypes for (A) Panicle weight (PAWT) and (B) Grain yield (GY).** The blue arrowhead line that passes through the origin shows higher mean performance of a genotype and the green dotted lines extending from the blue arrowhead line show the stability of the genotypes (the shorter the dotted line the higher the stability of the genotype).

most stable whereas G20 were the least stable genotypes (Fig 5A). Genotypes G244 and G118, for PALH, genotypes G261 and G5 for PAWD, genotypes G255 and G244 for PH were the most stable with a shorter vector from AEA. Genotype G313 was the least stable for both panicle length and width and genotype G153 was the least stable for PH (S3 Fig).

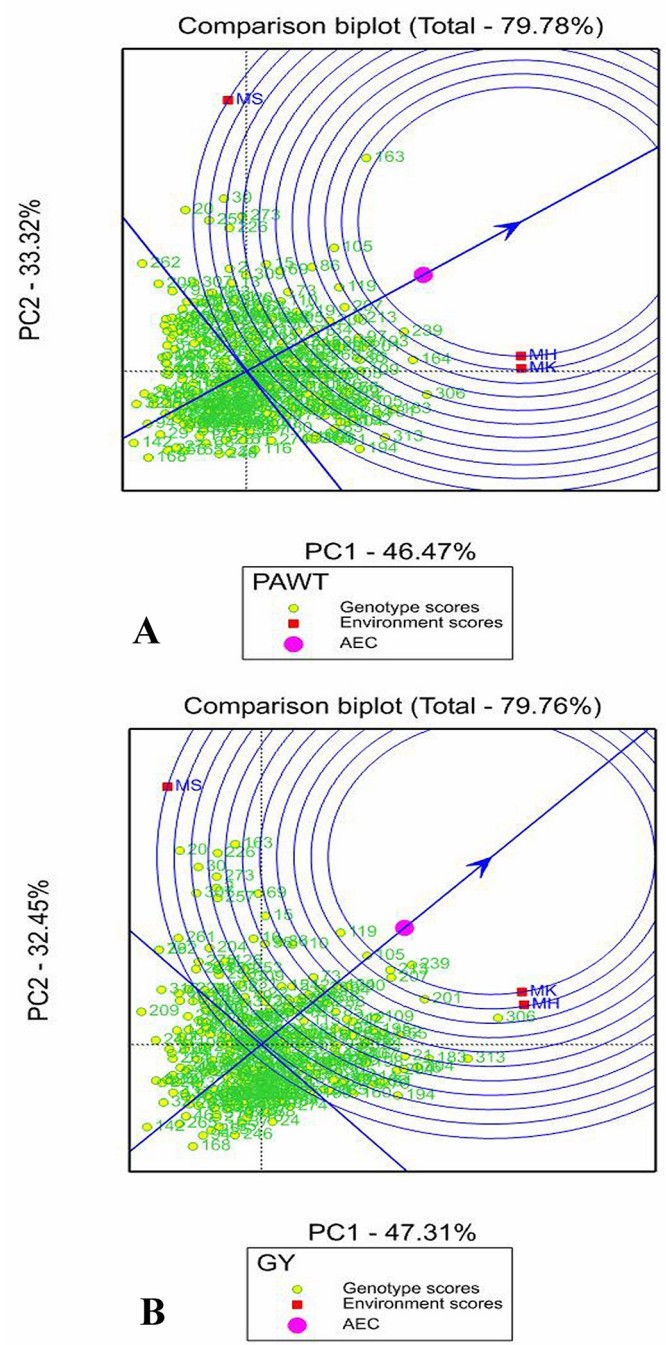

**Fig 6. Environment focus scaling GGE comparison biplot of the 324 sorghum genotypes ranking the three tested environments for (A) panicle weight (PAWT) and (B) grain yield (GY).** The concentric circles on the biplot show the distance of the environments from AEC and the biplot origin. The ideal environment is the one that is close to the center of the concentric circles.

**Evaluation of environments in comparison biplots.** Environment-focused scaling of comparison the GGE biplot shows AEA, AEC and concentric circles which helps to evaluate the tested environments. The concentric circles on the comparison GGE biplot graph (Figs 6 and S4) showed the distance of the environments to AEA, AEC and the biplot origin. The

ideal environment is the one that is close to the center of concentric circles. In this study, environment MK was the ideal environment (representative) for GY, PALH and PAWD as it is the closest to the center of concentric circles and having the smallest angle with AEA. The environments that were placed far from the comparison biplot origin indicated the discriminating ability of the environments and hence all three-tested environments had strong discriminating ability for all traits as they were placed far from the biplot origin.

## Correlation among traits

Significant positive and negative correlations were detected between traits studied (Fig 7). Grain yield showed that highly significant (P < 0.001) and high positive correlation with panicle weight (0.91). Significant (P < 0.01) and negative correlation were detected between grain yield vs. panicle length (-0.44) and panicle weight vs. panicle length (-0.50). Grain yield revealed a non-significant negative correlation with days to flowering (-0.12) and positive correlation with plant height (0.16).

## Cluster analysis

The dendrogram was generated from cluster analysis of the 324 sorghum genotypes based on the six traits. The cluster analysis grouped the genotypes in to five clusters (Fig 8). Cluster- IV

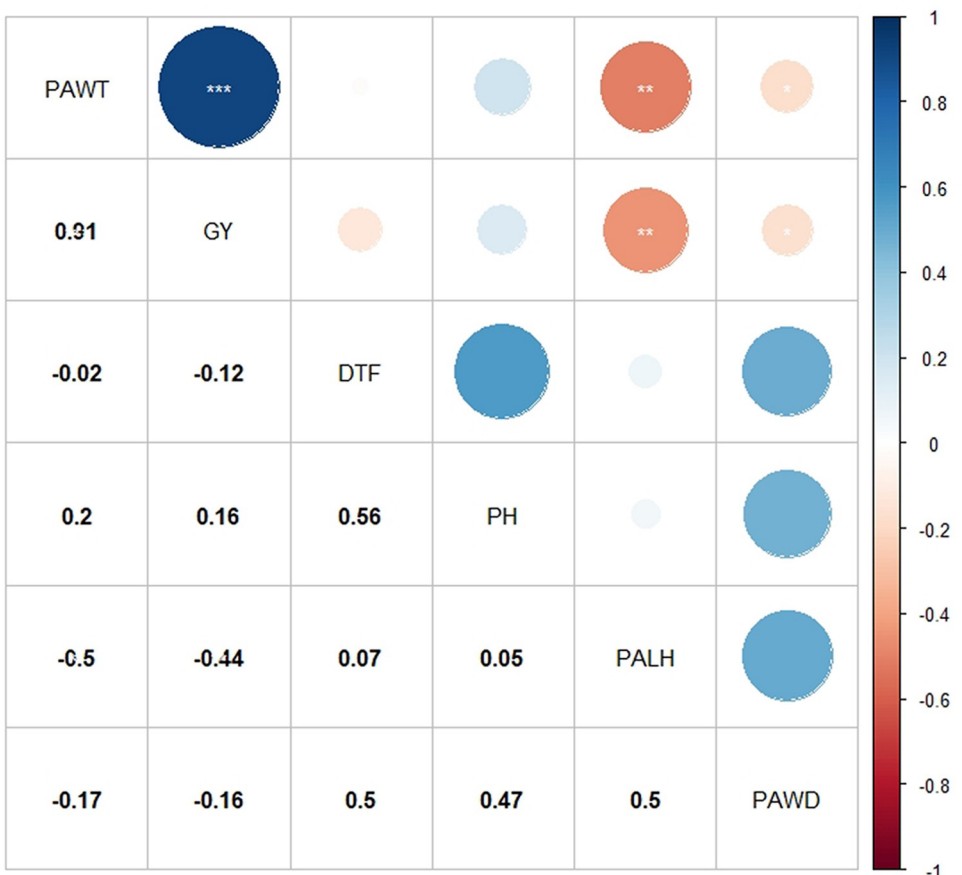

**Fig 7. Correlation coefficient and level of significant for grain yield and agronomic traits in 324 sorghum genotypes.** PAWT: Panicle weight, GY: Grain yield, DTF = Days to flowering, PH = Plant height, PALH = Panicle length, PAWD = Panicle width.

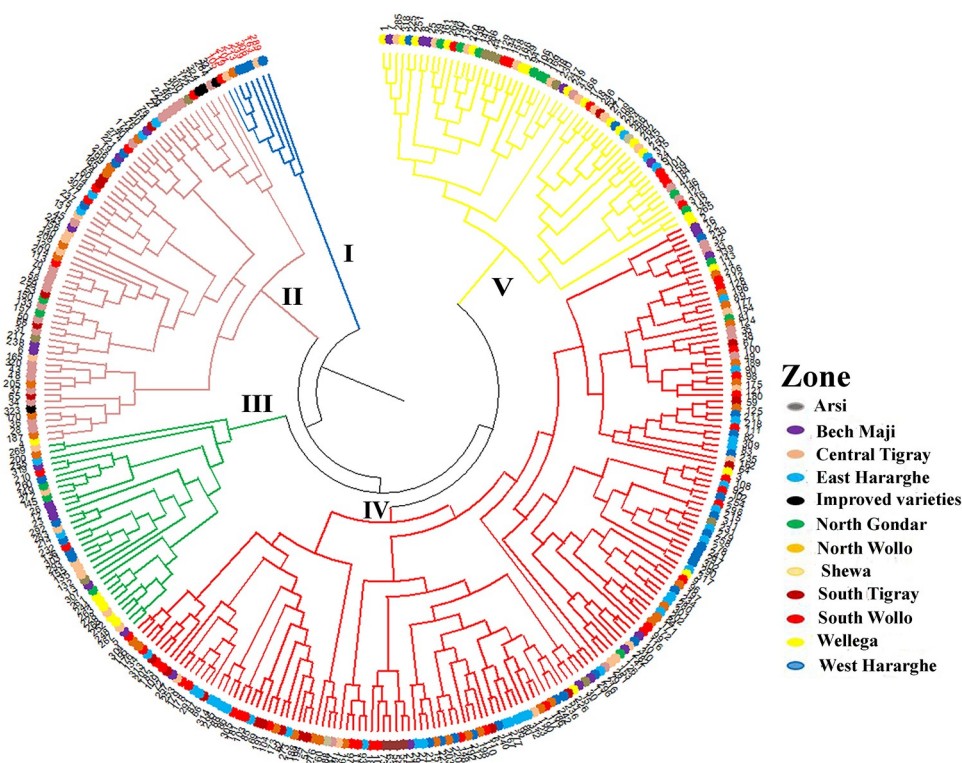

**Fig 8. Cluster analysis of the 324 sorghum genotypes using phenotypic data of the six studied traits revealing five major clusters (I to V).** Genotypes in cluster-I (red font) are high yielding.

was the largest one consisting of 158 genotypes, followed by Cluster-II that comprised 67 genotypes. Cluster-I, III and V contained 8, 33 and 58 genotypes, respectively. The cluster analyses showed good correspondence with the stability and the performance of genotypes obtained from the AMMI and GGE biplots. For instance, genotypes G163, G306, G239, G105, G119, and G201 in Cluster-I were among the high yielding genotypes according to the AMMI2 and GGE ranking biplots. Furthermore, all the four improved varieties were grouped together in Cluster-II. However, the cluster analysis did not clearly group genotypes based on the proximity of their geographical locations where they were initially collected. Significant numbers of genotypes from different regions were grouped together and genotypes from the same regions were placed under different clusters. For instance, nine genotypes originally collected from Shewa were grouped in Cluster-IV, while eight genotypes from the same zone were grouped in Cluster-III. Most of the genotypes from Central Tigray were grouped in Cluster-II whereas genotypes from North Wollo, East Harerge and West Harerge were grouped in Cluster-IV. South Wello, West Hararghe and Shewa had highly diverse genotypes that were distributed across all clusters (Fig 8).

## Discussion

Ethiopia is considered as the center of origin and diversity of sorghum [34,35], due to the presence of its wild and cultivated forms. In this study, combined and AMMI analysis of variance revealed a highly significant variation among the 324 sorghum genotypes (landraces and improved varieties) for the assessed traits. The high genetic variation revealed in this study and previous studies in sorghum landraces [36,37], indicated the presence of great opportunity to select and use the landraces for sorghum improvement programs.

Genotypic effect had higher contribution to the total variation in DTF (54.9%), PH (61.4%) and PAWD (53.9%) as compared to environment and G×E interaction effects. However, the effect of the environment was higher (33.6%) than the genotypic effect for variation in PAWT, and the effect of G×E interaction was higher (31.3%) than genotypic effect for variation in GY. The higher contribution of environment and G×E interaction to variation in grain yield were reported in sorghum [5,22] and other crops [38,39]. The significant effect of G×E interaction for the traits implies that different sorghum genotypes responded differently to variation in environmental conditions, leading to the necessity to identify and select environment specific genotypes. Higher contribution of G×E interaction as compared to genotype to variation in grain yield indicated the possible existence of different mega-environments across the testing environments [40,41]. The significant effect of the environment suggests the need to generate MET data that can lead to the identification of stable and top performing genotypes that have wide adaptation as well as for selection of genotypes with good adaptation to specific agro-ecology.

The variance due to genotype and G×E interaction helped to select the best genotypes for target traits, and in such cases, minimizing the impact of environmental main effects is important [10]. AMMI2 model was the best model to understand genotype stability and performance, genetic variation between genotypes and association with environments [42]. In the AMMI2 biplot, environments with low IPCA1 and IPCA2 scores (placed close to the origin) have high contribution to the stability of genotypes but with low contribution to the G×E interaction [14]. Thus, environment MH and MK were the top two contributors to the stability of genotypes in GY, PAWT, PALH and PH. Genotypes located far from the center and close to a given testing environment in AMMI2 biplot are considered well-adapted and high-performing in that environment [14]. For GY genotypes G20, G163, G226 and G30 were close to environment MS in this study, indicating their high performance and better adaptability are to this environment than the other two. On the other hand, genotypes G213, G306, G201 and G313 performed better in environment MK for GY. The difference in relative performance of genotypes at different environments is also a strong indicator of the existence of G×E interaction, and variation in environmental conditions such as temperature, rainfall, and soil type. This, therefore, suggests that environment-specific sorghum genotypes should be selected for different agro-ecologies and environmental conditions. High yielding genotypes under specific environments have been previously reported in sorghum [21,43] and barley [44].

Genotypes with low ASV and positioned close to the origin in AMMI2 biplot are generally regarded as highly stable [26]. In line with this, G70, G162 and G254 were identified as stable genotypes in grain yield. However, these genotypes had low mean grain yield, and should not be prioritized for use in breeding programs. In this study, GSI was used for selecting top ranking genotypes both in mean performance and stability [15,16], based on the ASV parameter (accounting for IPCA1 and IPCA2) and genotype mean ranking. This approach identified G148, G123, G110, G203 and G73 as stable genotypes with high grain yield across environments. Interestingly, these stable and high yielding genotypes were also top ranking in other farmer preferred traits such as high panicle weight (125, 122, 133, 118, 136 g) and plant height (309, 301, 308, 317, 279 cm), respectively. Moreover, these genotypes were collected from South Wello, West Hararghe and Shewa zones, which had highly diverse genotypes. Hence, these genotypes should be prioritized for use in sorghum breeding programs for further improvement in grain yield and other desirable traits. This method has been successfully used in other crops [40,45,46].

The GGE 'Which-Won-Where' biplot was used to identify top performing genotypes through interpreting the G×E interaction, MGE clustering and particular adaptation [10,11,14,22]. The genotypes which placed far from the biplot origin (vertex genotypes) are the

poorest or best performing in some or in all tested environments [47], which were more responsive to environmental change and are considered specifically adapted genotypes. Based on 'Which-Won-Where' biplot, the testing environments were grouped into two MGEs with different high performing genotypes for GY, PAWT and PAWD. For instance, for GY, MGE1 was represented by MK and MH environments containing G306, G313 and G194 as top grain yielding genotypes whereas MGE2 contained only environment MS where G163 and G20 were top performers in grain yield. This indicates that there were specific adaptations of genotypes to MGEs and hence positive exploitation of the G×E interaction [40]. The clustering of the target environments into meaningful MGEs and selecting different genotypes for different MGE is the best way to exploit the positive G×E interaction [33]. Such clustering of environments into MGEs and identification of top best performing genotypes adapted to a specific MGE have been reported in several crops [22,43,48,49].

The top ranking and stable genotypes can be identified by GGE ranking biplot through AEC [13]. In the present study, the ranking biplot AEC indicated genotypes G306, G239, G163, G201 and G213 as the top ranking in grain yield. However, the high yielding landraces such as G306, G163 and G201 were less stable landraces due to G×E interaction effect. Previous reports on forage and grain sorghum also showed that the high yielding genotypes are not necessarily the most stable [22,43]. A remarkable character of the GGE biplot graph is the visualization of genotypes that combine high mean performance and stability. The best genotypes could have larger projection on AEC (highest mean) along with shorter vector on AEA (high stability) [13,47,50]. Accordingly, genotypes G105, G213, G207, G239 and G119 were identified as high yielding and stable for grain yield. It implies that identification of ideal genotype through GGE biplot analysis is a suitable tool for detecting the most stable and the highest yielding genotypes. By using this method, several authors identified high yielding and stable genotypes in sorghum [21,43] and other crops including barley [40], soybean [41] and wheat [42].

The GGE biplot approach ranked genotypes G163, G239, G164, G105, G119, G207 and G213 on top for grain yield and panicle weight. Among these, genotypes G207 and G213 were identified as desirable for their stability and high mean panicle weight. The selection of the same genotypes for both GY and PAWT is mainly due to the positive association between the traits. On the other hand, different genotypes were identified as having high mean performance and stability for PALH, PAWD and PH. This study clearly indicated that a stable and high performing genotype in one trait does not necessarily mean that it combines stability and high performance in other related traits. This is largely the case because different traits are regulated by different genes and due to differential expression of genes among the genotypes as a response to environmental conditions, such as temperature variation and moisture stress. Similar results were reported in previous studies in sorghum [22,43] and wheat [51].

The AMMI analysis has been shown to be effective in capturing a large portion of the G×E interaction, by clearly separating the main and interaction effects using ANOVA and PCA [10]. GGE biplot is an effective statistical model for the identification of top ranking and stable genotypes across environments and best genotypes for adaptation to particular mega-environment [47]. The present study showed that the AMMI and GGE biplot models had similar results in the discriminating ability of the environments. Similarly, in both analyses, the environments were somehow similar in their discriminating ability as they were placed far from the biplot origin. However, somewhat different results were obtained in the contribution of the environments for genotype stability. The top ranking genotypes were similar in both AMMI and GGE biplot analysis. However, the ranking of genotype stability were somewhat different in the AMMI and GGE analysis. These results are in line with results obtained in

some other studies [42]. Such a difference is possible because of different statistical basis of IPCA in AMMI2 and PC in GGE biplot.

Grain yield showed that highly significant positive correlation with panicle weight (0.91). This is in agreement with previous studies on sorghum [52,53]. Hence, the positive correlation of grain yield with this trait showing possibility of simultaneous improvement of both traits through effective selection. Grain yield also revealed a positive, but non-significant correlation with plant height (0.16) and a negative correlation with days to flowering (-0.12). Amare et al. [53] also reported non-significant positive correlation between grain yield with plant height. Similar results of negative, or non-significant correlations between grain yield and days to flowering was reported by Akatwijuka et al. [54]. Negative correlation between grain yield and panicle length and width in this study were in contrast with other studies in sorghum [54,55]. This is mainly due to the variation in panicle shape and compactness of sorghum genotypes used in this study.

The present study revealed that the clustering patterns of genotypes were not largely a result of their geographic origin where they were originally collected in Ethiopia. The clustering of sorghum genotypes collected from the same geographical area in different clusters were also reported in previous studies [36,37,56]. This indicates genotypes in the same geographical region differ considerably in their agro morphological traits, indicating high genetic diversity in sorghum. The clustering of genotypes from different regions in the same groups is likely the results of gene flow across regions through market channels as well as a gradual exchange of seeds among farmers. The four improved varieties included in this study were grouped in the same cluster. These varieties were early maturing and short in height. Similar clustering of improved varieties from Ethiopian sorghum landraces were reported in previous studies [37,56]. The clustering of the best genotypes for grain yield identified through AMMI and GGE biplot analyses in the same group suggests that such genotypes were selected for the same traits (mainly grain yield) that led them to be more similar but showed higher differentiation from the other genotypes.

## Conclusions

This study determined G×E interaction effect, stability of genotypes and representativeness and discriminating ability of environments for days to flowering, plant height, panicle length, panicle width, panicle weight and yield in diverse sorghum genotypes grown in Ethiopia. Grain yield and panicle weight were highly affected by environmental variation and genotype by environment interaction whereas days to flowering, panicle length, panicle width and plant height were mainly affected by genotypic variation. The results obtained in this study clearly showed that the sorghum landraces are excellent genetic resources that contain high variation in grain yield and farmer-preferred traits such as plant height, which should be utilized for developing new high yielding cultivars with various desirable traits. The AMMI and GGE biplot models are effective in visualizing the G×E interaction and identifying stable and high performing genotypes. Among the 324 genotypes, G148, G123, G110, G203 and G73 were the best in terms of providing high and stable grain yield in combination with farmer-preferred traits. Among the studied populations, South Wello, West Hararghe and Shewa zones had highly diverse genotypes and hence these areas can be considered as a potential area for screening high yielding and other agronomic traits. Environment MK was the most representative environment whereas environment MS was the most discriminating, and hence should be used for capturing superior genotypes and for identification of high yielding genotypes for adaptation to specific agro-ecologies.

## Supporting information

**S1 Fig. AMMI2 biplot of 324 sorghum genotypes and three environments for (A) Panicle length (PAWT), (C) Panicle width (PAWD) and (B) Plant height (PH).** Genotypes placed close to a given environment, had top performance in that environment. Each vector shows the discrimination power of the environment (the longer the vector the more discrimination power that environment has).
(TIF)

**S2 Fig. Which-Won-Where polygon view of GGE scatter biplot of the 324 sorghum genotypes for (A) Panicle length (PAWT), (C) Panicle width (PAWD) and (B) Plant height (PH), showing genotypes with best performance in each environment and the mega environments (MGEs).** The vertex genotypes on convex hull (Polygon) are the best in each mega environment for the corresponding trait.
(TIF)

**S3 Fig. Genotype focus scaling of GGE biplot showing stability and mean ranking of 324 sorghum genotypes for (A) Panicle length (PAWT), (C) Panicle width (PAWD) and (B) Plant height (PH).** The blue arrowhead line that passes through the origin shows higher mean performance of a genotype and the green dotted lines extending from the blue arrowhead line show the stability of the genotypes (the shorter the dotted line the higher the stability of the genotype).
(TIF)

**S4 Fig. Environment focus scaling GGE comparison biplot of 324 sorghum genotypes to rank the three tested environments for (A) Panicle length (PAWT), (C) Panicle width (PAWD) and (B) Plant height (PH).** The concentric circles on the biplot show the distance of the environments from AEC and the biplot origin. The ideal environment is the one that is close to the center of the concentric circles.
(TIF)

**S1 Table. Genotype code and passport data of 324 sorghum genotypes.**
(XLSX)

**S2 Table. The top 10 stable genotypes ranked by AMMI stability value (ASV) for Grain yield (GY), Panicle weight (PAWT), Panicle length (PALH), Panicle width.**
Gen = genotype; IPCAg1 = first interaction principal component axis scores for genotype; IPCAg2 = second interaction principal component axis scores for genotype; ASV = AMMI stability value; rASV = rank of AMMI stability value; GSI = genotype selection index.
(XLSX)

## Acknowledgments

The authors thank the Swedish International Development Cooperation Agency (Sida) for financing this research. The authors also thank Melkassa Agricultural Research Center and Ethiopia Biodiversity Institute for providing sorghum germplasm used in this study. We are grateful to Melkassa, Mehoni and Mieso (a sub center of Melkassa) agricultural research centers for allowing us to use their experimental stations and for providing tools and assistance in the field trial activities. We also thank the Institute of Biotechnology, Addis Ababa University for technical support during the course of the study. The authors thank Professor Gary Wessel, Brown University, United States of America for editing the language.

## Author Contributions

**Conceptualization:** Muluken Enyew, Tileye Feyissa, Mulatu Geleta, Kassahun Tesfaye, Cecilia Hammenhag, Anders S. Carlsson.

**Data curation:** Muluken Enyew.

**Formal analysis:** Muluken Enyew.

**Funding acquisition:** Tileye Feyissa, Mulatu Geleta, Kassahun Tesfaye, Cecilia Hammenhag.

**Investigation:** Muluken Enyew.

**Methodology:** Muluken Enyew, Tileye Feyissa, Mulatu Geleta, Kassahun Tesfaye, Cecilia Hammenhag.

**Project administration:** Tileye Feyissa, Cecilia Hammenhag, Anders S. Carlsson.

**Supervision:** Tileye Feyissa, Mulatu Geleta, Kassahun Tesfaye, Cecilia Hammenhag, Anders S. Carlsson.

**Validation:** Muluken Enyew, Anders S. Carlsson.

**Visualization:** Muluken Enyew, Anders S. Carlsson.

**Writing – original draft:** Muluken Enyew.

**Writing – review & editing:** Muluken Enyew, Tileye Feyissa, Mulatu Geleta, Kassahun Tesfaye, Cecilia Hammenhag, Anders S. Carlsson.

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
