## [Decision Letter · Decision Letter 0]

1 Jul 2021

PONE-D-21-10761

Genotype by environment interaction, correlation, AMMI, GGE biplot and cluster analysis for Grain yield and other agronomic traits in Sorghum (Sorghum bicolor L. Moench)

PLOS ONE

Dear Dr. Enyew,

Thank you for submitting your manuscript to PLOS ONE. After careful consideration, we feel that it has merit but does not fully meet PLOS ONE’s publication criteria as it currently stands. Therefore, we invite you to submit a revised version of the manuscript that addresses the points raised during the review process.

We look forward to receiving your revised manuscript.

Kind regards,

Mehdi Rahimi, Ph.D.

Academic Editor

PLOS ONE

Journal Requirements:

3. Please ensure that you refer to Figure 2 in your text as, if accepted, production will need this reference to link the reader to the figure.

4. We note that Figure 1 in your submission contain map images which may be copyrighted. All PLOS content is published under the Creative Commons Attribution License (CC BY 4.0), which means that the manuscript, images, and Supporting Information files will be freely available online, and any third party is permitted to access, download, copy, distribute, and use these materials in any way, even commercially, with proper attribution. For these reasons, we cannot publish previously copyrighted maps or satellite images created using proprietary data, such as Google software (Google Maps, Street View, and Earth). For more information, see our copyright guidelines: http://journals.plos.org/plosone/s/licenses-and-copyright.

You may seek permission from the original copyright holder of Figure 1 to publish the content specifically under the CC BY 4.0 license. 

If you are unable to obtain permission from the original copyright holder to publish these figures under the CC BY 4.0 license or if the copyright holder’s requirements are incompatible with the CC BY 4.0 license, please either i) remove the figure or ii) supply a replacement figure that complies with the CC BY 4.0 license. Please check copyright information on all replacement figures and update the figure caption with source information. If applicable, please specify in the figure caption text when a figure is similar but not identical to the original image and is therefore for illustrative purposes only.

Reviewers' comments:

Reviewer's Responses to Questions

**Comments to the Author**

1. Is the manuscript technically sound, and do the data support the conclusions?

Reviewer #1: Yes

2. Has the statistical analysis been performed appropriately and rigorously? 

Reviewer #1: Yes

3. Have the authors made all data underlying the findings in their manuscript fully available?

Reviewer #1: Yes

4. Is the manuscript presented in an intelligible fashion and written in standard English?

Reviewer #1: Yes

5. Review Comments to the Author

Reviewer #1: The manuscript by Enyew et al. seeks to address the complications associated with G x E interactions in sorghum, and reports of elite cultivars. The research has been rigorously performed. However, my few comments are as follows;

1. Sentences in the abstract should be carefully looked at, and must be concise as possible.

2. If possible, authors should recheck for grammatical errors. For instance line 53,.... be "used"; 65, ... it's grain yield "varies", and many more.

3.Authors should ensure consistency of the tables and make sure they meet the requirements of the journal.

6. PLOS authors have the option to publish the peer review history of their article (what does this mean?). If published, this will include your full peer review and any attached files.

Reviewer #1: No

---

## [Author Response · Author response to Decision Letter 0]

27 Jul 2021

A rebuttal letter

Journal Requirements: 

 We have thoroughly checked and edited the manuscript based on the required style of PLOS ONE. Thank you for the comments.

We have thoroughly read the manuscript and grammatical and editorial errors have been corrected in “track changes” mode. Additionally, the language usage, spelling, and grammar have also been checked by native English language speaker (Professor Gary Wessel, Brown University, USA) whom we duly acknowledged in the Acknowledgments section of this manuscript. Thank you for the comments. 

3. Please ensure that you refer to Figure 2 in your text as, if accepted, production will need this reference to link the reader to the figure.

 The comment is accepted and the figure is now cited in the body of the revised manuscript. 

4. We note that Figure 1 in your submission contain map images which may be copyrighted. All PLOS content is published under the Creative Commons Attribution License (CC BY 4.0), which means that the manuscript, images, and Supporting Information files will be freely available online, and any third party is permitted to access, download, copy, distribute, and use these materials in any way, even commercially, with proper attribution. For these reasons, we cannot publish previously copyrighted maps or satellite images created using proprietary data, such as Google software (Google Maps, Street View, and Earth). 

Figure 1 was constructed using ArcGIS software. First, the shape file of the country and the regions were downloaded from the DVIA-GIS (https://www.diva-gis.org/gdata), which is a freely available computer program for mapping and for downloading and analyses of geographic data of any country. Then, the ArcGIS software was used to generate Figure 1. Hence, the Figure is not copyrighted. 

Response to Reviewer Comments

We thank the reviewer for their thoroughly reading the manuscript and for their thoughtful comments and constructive suggestions, which helped us to improve the quality of this manuscript. We have accommodated all the comments and accepted the suggestions. We have revised the manuscript using the track changes mode in MS Word. 

Reviewer #1: The manuscript by Enyew et al. seeks to address the complications associated with G x E interactions in sorghum, and reports of elite cultivars. The research has been rigorously performed. However, my few comments are as follows;

1. Sentences in the abstract should be carefully looked at, and must be concise as possible.

The comment is accepted and the abstract is carefully revised. 

2. If possible, authors should recheck for grammatical errors. For instance line 53,.... be "used"; 65, ... it's grain yield "varies", and many more.

We have carefully checked and revised the manuscript and corrected grammatical and editorial errors. The language usage, spelling, and grammar have also been checked and edited by native English language speaker (Professor Gary Wessel, from Brown University, USA). Thank you for the comments.

3. Authors should ensure consistency of the tables and make sure they meet the requirements of the journal.

We have checked the consistency of the tables and rearranged them according to the required style of PLOS ONE.

---

## [Decision Letter · Decision Letter 1]

22 Sep 2021

Genotype by environment interaction, correlation, AMMI, GGE biplot and cluster analysis for grain yield and other agronomic traits in sorghum (Sorghum bicolor L. Moench)

PONE-D-21-10761R1

Dear Dr. Enyew,

We’re pleased to inform you that your manuscript has been judged scientifically suitable for publication and will be formally accepted for publication once it meets all outstanding technical requirements.

Kind regards,

Mehdi Rahimi, Ph.D.

Academic Editor

PLOS ONE

Additional Editor Comments (optional):

Reviewers' comments:

Reviewer's Responses to Questions

**Comments to the Author**

1. If the authors have adequately addressed your comments raised in a previous round of review and you feel that this manuscript is now acceptable for publication, you may indicate that here to bypass the “Comments to the Author” section, enter your conflict of interest statement in the “Confidential to Editor” section, and submit your "Accept" recommendation.

Reviewer #1: All comments have been addressed

2. Is the manuscript technically sound, and do the data support the conclusions?

Reviewer #1: Yes

3. Has the statistical analysis been performed appropriately and rigorously? 

Reviewer #1: Yes

4. Have the authors made all data underlying the findings in their manuscript fully available?

Reviewer #1: Yes

5. Is the manuscript presented in an intelligible fashion and written in standard English?

Reviewer #1: Yes

6. Review Comments to the Author

Reviewer #1: All reviewer comments have been addressed and manuscript has been well prepared. It is therefore fit to be accepted for publication.

7. PLOS authors have the option to publish the peer review history of their article (what does this mean?). If published, this will include your full peer review and any attached files.

Reviewer #1: No

---

## [Editor Report · Acceptance letter]

27 Sep 2021

PONE-D-21-10761R1 

Genotype by environment interaction, correlation, AMMI, GGE biplot and cluster analysis for grain yield and other agronomic traits in sorghum (Sorghum bicolor L. Moench) 

Dear Dr. Enyew:

I'm pleased to inform you that your manuscript has been deemed suitable for publication in PLOS ONE. Congratulations! Your manuscript is now with our production department. 

Kind regards, 

on behalf of

Dr. Mehdi Rahimi 

Academic Editor

PLOS ONE